# Comparison of Barrier Surveillance Algorithms for Directional Sensors and UAVs

**DOI:** 10.3390/s24144490

**Published:** 2024-07-11

**Authors:** Bertalan Darázs, Márk Bukovinszki, Balázs Kósa, Viktor Remeli, Viktor Tihanyi

**Affiliations:** Széchenyi István University, 9026 Győr, Hungary; darazs.bertalan@techtra.hu (B.D.); bukovinszki.mark@techtra.hu (M.B.); remeli.viktor@techtra.hu (V.R.); tihanyi.viktor@techtra.hu (V.T.)

**Keywords:** sensors, coverage, tracking, cost minimization, linear sum assignment, reinforcement learning

## Abstract

Border surveillance and the monitoring of critical infrastructure are essential components of regional and industrial security. In this paper, our purpose is to study the intricate nature of surveillance methods used by hybrid monitoring systems utilizing Pan–Tilt–Zoom (PTZ) cameras, modeled as directional sensors, and UAVs. We aim to accomplish three occasionally conflicting goals. Firstly, at any given moment we want to detect as many intruders as possible with special attention to newly arriving trespassers. Secondly, we consider it equally important to observe the temporal movement and behavior of each intruder group as accurately as possible. Furthermore, in addition to these objectives, we also seek to minimize the cost of sensor usage associated with surveillance. During the research, we developed and analyzed several interrelated, increasingly complex algorithms. By leveraging RL methods we also gave the system the chance to find the optimal solution on its own. As a result we have gained valuable insights into how various components of these algorithms are interconnected and coordinate. Building upon these observations, we managed to develop an efficient algorithm that takes into account all three criteria mentioned above.

## 1. Introduction

Border surveillance and the monitoring of critical infrastructure are essential components of regional and industrial security, as they help in the detection and prevention of unauthorized crossings and activities. Traditional methods of border surveillance predominantly rely on human resources. This involves the establishment of checkpoints along the perimeters of the secured zones where personnel can monitor for any unauthorized activities. Additionally, at large industrial establishments and national borders, regular patrols are organized at predetermined intervals. However, monitoring mid-sized to large areas requires significant human involvement which can be practically ineffective and hard to organize.

To address these limitations, there has been a shift towards the integration of intelligent technologies in border surveillance operations. Recent advancements have led to systems that utilize Wireless Sensor Networks (WSN) for remote monitoring purposes [1,2]. Authors of [3], have created a smart WSN system that alerts a human controller if an intruder is detected using PIR sensors and camera object detection. Furthermore, the adoption of Unmanned Aerial Vehicles (UAVs) has significantly enhanced surveillance capabilities offering improved outcomes in surveillance [4] and effective tracking of border intruders [5].

Some researchers have explored multi-modal sensor systems that incorporate cameras, UAVs, Unmanned Ground Vehicles (UGVs), and underground sensors in a unified framework [6]. While such systems offer minimal false alarm rates and comprehensive coverage, they are constrained by high costs and environmental limitations. On the other hand, efforts to deploy static sensors efficiently have focused on optimizing sensor placement to minimize coverage gaps and guarantee intruder detection, although requiring a high density of sensors for effectiveness [2].

In real-world applications the effective coverage and tracking of targets is a complex challenge. To continuously monitor the targets, researchers devised various strategies. One approach uses Integer Linear Programming (ILP) and a hierarchical approach to cover a set of static targets with Pan–Tilt–Zoom (PTZ) cameras [7]. For targets that move, adaptive strategies have been developed to reconfigure camera positions ensuring that the highest number of targets are covered at any given time [8].

The use of Reinforcement Learning (RL) has emerged as a promising approach to dynamically configure surveillance systems with prior knowledge of target positions [9]. In fact, most works focus on the problem where the target positions are known. While many studies assume the positions of targets are known, a few, including a follow-up to the aforementioned RL study, deal with the problem where the target positions are unknown offering a more realistic simulation of surveillance scenarios [10].

In this paper, we address the problem of mobile target coverage in scenarios where both the number and positions of the targets are unknown. Our purpose is to study the intricate nature of surveillance methods used by hybrid monitoring systems utilizing Pan–Tilt–Zoom (PTZ) cameras, modeled as directional sensors, and UAVs, a concept not extensively explored in the existing literature. We analyze several algorithms built upon each other. When developing these methods, we aim to accomplish three occasionally conflicting goals. Firstly, at any given moment we want to detect as many intruders as possible with special attention to newly arriving trespassers. Secondly, we consider it equally important to observe the temporal movement and behavior of each intruder group as accurately as possible. Furthermore, in addition to these objectives, we also seek to minimize the cost of sensor usage associated with surveillance.

Initially, we focused on methods only using cameras. In the basic algorithm each camera progresses through predetermined points one after the other in such a way that it observes each individual point over a predefined time interval. This heuristic is obviously designed to attain the first goal. However, it is not obvious, in order to address the second criterion, how tracking could be incorporated in such a way that the associated costs are minimized. Additionally, it is also an interesting question how the new algorithm performs in comparison to the previous one.

Along similar lines of inquiry, we developed a total of four camera-based algorithms where the last one, unlike the previous ones, is based on RL. It turned out that the learned behavior of this method also helped to select more optimal parameters for the previous three algorithms.

In a similar manner, we also studied algorithms that exclusively utilize UAVs. However, the dominance of camera-based algorithms quickly became clear in this case.

Finally, we combined the camera and UAV algorithms developed in the previous steps, retaining those elements that proved useful during empirical investigations.

Our experiments were performed within a 2D simulation environment. We used two different settings. With the first we aimed to represent a more abstract scheme in which intruders may appear at any given point of the observed area with equal probability. On the other hand, the second variant modeled a scenario with several aspects taken from real-world situations.

To summarize our results, we have developed a barrier surveillance algorithm employing cameras and UAVs which detects and then follows intruder groups aiming to beat the system. During the execution of the algorithm we aim to minimize the cost of sensor usage by solving appropriate instances of the linear sum assignment problem [11]. In the process of the development of this final method, we examined seven intermediate algorithms which provided insight into how certain aspects of such procedures are interconnected and function together.

This paper is organized as follows. In Section 2 the related works are presented. In Section 3 the details of the 2D simulation are described and the indicators used to assess the performance of the algorithms are introduced. Next, in Section 4 the algorithms are explained and compared to each other. Since the RL-based method is entirely different from the rest of the techniques, it is described separately in Section 5. Finally, in Section 6 we conclude our results.

## 2. Related Work

There are several categories and subcategories of the coverage problem such as area coverage, target coverage [12], barrier coverage [13], quality- or priority-based target coverage [14], etc.

Primarily, we wanted to solve a variant of the target coverage task. Specifically, in any given time step, we aimed to cover as many targets as possible with the minimum number of sensors so that the unused sensors can be utilized to monitor the area and detect new intruders. The Maximum Coverage with Minimum Sensors (MCMS) problem was introduced in [15] for directional sensors. However, in this and the subsequent works that offer solutions to various extended versions of the original problem the targets are static, thus it is enough to run the optimization algorithm only once. In our case, the targets continuously change their positions, so the optimization should be accomplished at every time step.

Another important difference is that when the costs in a sensor network are to be optimized, in most of the papers, since sensors are abundantly available, the focus is on putting unnecessary sensors to sleep and activating them only when needed [15,16,17]. In our case though, the number of sensors was limited, as it will become clear later, they could not provide even one-barrier coverage, so it was not permissible to disable any of them. Thus, instead of disabling as many sensors as was possible, at each time step we aimed to minimize the magnitude of rotation angles necessary to adapt to the new situation. This approach was also taken in [18]. In the case of UAVs, however, we also wanted to minimize the distance required to complete the task, just as most related articles did [19].

A further significant difference is that in our system, the cooperation of PTZ cameras and UAVs had to be solved, while in most works, the sensors in the network belonged to one type.

All in all, we did not find any scientific publication that aimed to solve the same complex task as we intended to address. From this perspective, we found [9] to be the closest work to ours. The authors even compare the efficiency of their solution with that of an algorithm that solves the MCMS problem at every time step. However, in [9] reinforcement learning was used, resulting in a method that functions as a black box regarding the decisions made. Instead, we aimed for a solution that operates in a way understandable to humans. We used our reinforcement learning-based algorithm only to help determine the correct values of the parameters used by our method based on more “traditional” optimization approaches. This also excluded the use of other approaches that otherwise appear in the literature, such as learning automata [20] and genetic and ant-colony algorithms [21,22].

However, when developing our patrolling algorithm relying only on UAVs, we essentially implemented an existing method [23]. The details are given in Section 4.2.1.

Furthermore, in formulating the performance indicators measuring the efficiency of the algorithms, in some cases we relied on existing metrics [24]. Again, the relevant information can be found in Section 3.2.

Finally, in designing the 2D simulations we used the result of [25] providing a method for deciding whether an open belt region is one-barrier covered or not (Section 4.1.1). Note that the algorithm is developed for omnidirectional sensors, so our aim was not to create one-barrier coverage, rather, it was to justify the strategy used for selecting the positions of the cameras.

## 3. Methodology

### 3.1. Simulation

To be able to test the efficiency of our algorithms, we used two-dimensional simulations in which we could model how the sensor system operates as well as the movement and behavior of intruders. As a framework we relied on an application developed in the Department of Automotive Technologies at Budapest University of Technology and Economics.

#### 3.1.1. Simulation of the Environment

Our goal was twofold. On the one hand, we aimed to test the algorithms in a more general, theoretical environment. On the other hand, we also wanted to assess a more realistic scenario with some central characteristics of practical applications. These latter endeavors were backed by some experts in the field. Later, it will be useful to refer to these environments using short names, thus in the sequel these scenes will be called ABSTRACT and REAL, respectively.

In both cases, the width of the playing field was 1920 units, whereas its height was 1080 units. The border, which we aimed to monitor, was represented by a horizontal line in the middle of the playing field.

One of the most important aspects we wanted to model in the ABSTRACT environment was that intruders could occur at any point in space with equal probability. Intruders typically move along paths, thus we employed a dense, regular network of paths here (see Figure 1). In the REAL case the walkway system was less dense and more irregular, aiming to represent that intruders in reality do not generally appear anywhere, but rather attempt to cross the border by following several, yet a limited number of known paths (Figure 2).

#### 3.1.2. Simulation of the Intruders

Intruders usually do not act alone but rather organize into groups when attempting to traverse the border. Furthermore, it is quite commonly observed that initially larger teams eventually split into smaller groups.

To represent this behavior, rather than spawning individual intruders, we create groups of them. These groups are further organized into pools. All members of a pool start their journey from the same entry point. Members of a group always move together, while groups belonging to a pool can split apart over time. A group may contain 5–10 intruders and a pool consists of 1–3 groups. At a single spawning 5–11 pools are created in the ABSTRACT case and 3 pools in the REAL setting. Spawning takes place every 15–30 s. The exact numbers within the aforementioned intervals are selected randomly according to the corresponding discrete uniform distribution.

Each group has a preference for choosing the left or right direction when they arrive at a junction point. More concretely, in the simulation each group is assigned an integer *c* in its creation, where *c* is in [1,10]. Then, in every decision-making situation they go to the left with c/10 probability.

#### 3.1.3. Recognition of Intruder Groups

Sensors detect individuals, not groups. Since we are to track the movement of groups the system should be able to organize these individual detections into groups. For the sake of simplicity, each such group is represented by a circle.

Formally, the following problem should be solved: for given nodes on a two-dimensional plane and for a fixed radius we aim to find the minimal number of circles that cover all of the nodes. To solve this problem we applied a greedy heuristic introduced in [26]. The algorithm works as follows (see Figure 3 for a graphical representation of the steps).

At the first step, for each node pair, whose Euclidean distance is less than or equal to 2∗r, we consider the circle or those two circles which contain these two nodes on their outer edge. (If the distance between these two nodes is equal to 2∗r, then there is only one such circle, otherwise there are two.) Here, *r* denotes the given radius.In the second step, for each node that was not included in any of the circles constructed in the previous step, we draw a circle around this point with radius *r*.In the third step, for each circle *C* we assign those nodes that are inside or on the boundary of this circle. Denote as S(C) this assigned node-set.In the last step, first we take the circle with the maximum number of assigned nodes and add it to the result, i.e., to the set of covering circles which we aim to find. Denote as $C_{M}$ this circle. For each of the remaining circles *C* the nodes in $S(C_{M})$ are removed from S(C). This process is repeated until there is a circle $C^{\prime }$ to which $S(C^{\prime })$ is not empty.

It is easy to see that this algorithm runs in $O(n^{3})$ time, where *n* is the number of nodes.

At each time step of the simulation after receiving the detection of individuals, we always run this algorithm. Thus, in what follows, whenever we speak about an intruder group in the context of the simulation, we always mean one of the circles returned by the algorithm.

#### 3.1.4. Simulation of the Cameras

In the simulation we consider cameras as directional sensors, thus they are modeled as circular sector-shaped polygons that can rotate around their center [12]. We work with PTZ (Pan/Tilt/Zoom) cameras which are characterized by static and dynamic properties.

The static properties of the cameras are given by configuration parameters which include the vertical and horizontal field of view, the rotation speed around the axis, and the detection range. Since we run two-dimensional simulations, only the horizontal field of view has practical importance in our case.

In a given time step the dynamic state of a camera is defined by its pan, tilt, and zoom values. The pan value indicates the horizontal rotation around the axis, while the tilt value the vertical. Again, the tilt value is not taken into account, thus only the pan and zoom values have significance. In the simulation the detection range of a camera changes proportionally with the zoom. A higher zoom value entails a longer detection range and narrower horizontal field of view.

The maximum rotation speed for the cameras is taken to be 90 degrees per second. The detection range is between 176 and 220 units of distance depending on the zoom (1–1.25), and the horizontal field of view can vary between 60 and 48 degrees.

When we want to turn a camera towards an intruder group, more precisely towards the center of the circle representing this intruder group, the coordinates should be transformed to a pan value or orientation. This calculation is straightforward and can be achieved by using the two-argument arctangent function [27].

#### 3.1.5. Detection of Intruders

In order to decide when an intruder is detected by a camera, we rely on the Target in Sector (TIS) test [12,15]. Here, target stands for the intruder, and sector refers to the circular sector covered by the camera.

According to this test, two conditions must be met for a camera to detect an intruder. Firstly, the intruder must be within the detection range:(1)$${∥v∥}_{2}\leq R_{c}$$
Here, v is the vector pointing from the camera to the intruder and $R_{c}$ denotes the maximum detection range of the camera.

Secondly, the intruder must be within the field of view of the camera given its current orientation:(2)$$d^{T}v\geq {∥v∥}_{2}\cos\left(\frac{\theta }{2}\right),$$
where d is the unit vector with the same direction as the orientation of the camera and θ denotes the horizontal field of view.

Note that this is a *binary detection model*, which means that if the aforementioned two conditions are met, the camera is guaranteed to detect the intruder.

#### 3.1.6. Simulation of UAVs

In the simulation UAVs are modeled as circles that can move in any direction in the two-dimensional space. In a similar way as in the case of cameras, UAVs can be characterized by both static and dynamic properties. Among the static properties we only consider the maximum speed, which is configured to be 30 units per time step. Among the dynamic properties, only the current position has practical significance from the perspective of the simulation.

An intruder is considered to be detected by a UAV if the distance between the center of the circle representing the UAV and the intruder is less than the detection distance of the UAV. This is configured to be 100 units for each UAV.

### 3.2. Performance Indicators

To be able to assess the performance of our algorithms, we introduce five different indicators. The first three describe the efficiency from different aspects, while the last two quantify the cost of camera and UAV usage respectively.

The first indicator expresses the tracking efficiency of an algorithm. For each intruder intr we consider the length of time when intr was detected by at least one sensor and divide this number by the whole length of time that intr spent on the playing field. Then, we take the average of these ratios. We refer to the result of this computation as the *average tracking ratio*.

In contrast to the prior figure, which is a temporal quantity, the second indicator is a screenshot taken at every time step of the simulation. At each time step, we calculate the ratio of intruders detected by any of the sensors to the total number of intruders currently present on the playing field. Next, as in the previous case, the average of these ratios is taken. The final result is called *average coverage* in the sequel.

At first glance, it may seem that the two indicators ultimately result in the same value. To understand the difference, consider the following toy example. Intruder A is on the playing field in the first two time steps, and is detected in both time steps:(1,True),(2,True).
Furthermore, take Intruder B with
(1,True),(2,True),(3,True),(4,False).
Then, the average tracking ratio is (1+0.75)/2=0.875, while the average coverage is (1+1+1+0)/4=0.75.

If only Intruder B was taken into account, then the two indicators would be the same. Meanwhile, the two seconds that Intruder A spent on the playing field affect the two indicators in a different way.

A closer look at Figure 1 and Figure 2 reveals that the cameras do not cover the whole playing field. Consequently, the previous indicators may provide a pessimistic approximation of the performance of a method. To counterbalance this effect we also compute these figures for a constrained dataset in which we only consider an intruder when it can be detected by at least one camera. We will refer to this area as the *camera-accessible region*.

From a practical perspective, it is crucial to keep track of the number of intruders who were not detected at all while they were on the playing ground. The third indicator gives the ratio of these intruders compared to all intruders who have ever entered the playing field in the given simulation. It is referred to as the *ratio of non-observed intruders*, which will be abbreviated sometimes as the *non-observed ratio*. Note that in this case, for clear reasons, the analysis should not be narrowed down to the camera-accessible region.

The fourth indicator, the cost of camera usage, is based on the magnitude of the camera rotation. In order to more vividly express the capacity utilization of a camera, in each time step we take this magnitude and divide it by the maximum rotation capacity, which is 90 in our case (Section 3.1). Again, in the final step the average of these values is taken. We refer to the result as the *average camera capacity utilization* or, shortened, *average camera utilization*.

In a similar manner as for UAVs, in each time step the traveled distance is taken to be the cost which is divided by the maximum capacity (30).

Note that the metrics defined in [24]—camera move time, target leakage and target coverage frequency—are similar in nature to the average camera capacity utilization, the ratio of non-observed intruders and the average tracking ratio indicators, respectively.

## 4. Algorithms

### 4.1. Camera Algorithms

#### 4.1.1. Scanning

Normally, the simplest method used in practice for intruder detection is to define a fixed set of observation points around a camera and direct it to these points one after the other, changing the position at predefined time intervals. For such an algorithm to work efficiently, it is crucial to select the most appropriate observation points and the proper amount of time spent at each point by the camera. Usually, these decisions are dictated by the circumstances of the particular environment where the sensor system operates.

In our simulations instead of specifying fixed points, we select a set of orientation zoom value pairs for each camera. For the sake of simplicity, the zoom values are taken to be the same.

##### Placement of the Cameras

Recall from the previous section that the border that we aim to monitor is represented by a straight, horizontal line running in the middle of the playing field. Let us also remember that the ABSTRACT scenario is meant to represent a homogeneous, uniform environment constructed in such a way that a given intruder may appear at each junction point of the dense network of the paths with the same probability. Intuitively, if one is to monitor a horizontal line in this setting, then the cameras should be placed with the least possible overlapping in such a way that they cover the largest possible horizontal area.

In the simulation we have four cameras, and $8*R_{c}$ is less than the width of the playing field, where $R_{c}$ denotes the maximum detection range of a camera. In other words, this means that if the cameras are placed along the horizontal line in such a way that they divide this line into five segments with equal lengths, then there will not be any overlapping among them. However, if the outermost cameras reached the two edges of the playing field and there was overlap between neighboring cameras, this arrangement would provide one-barrier coverage for omnidirectional cameras [25], further justifying the validity of this placement strategy. Thus, in accordance with the previous intuition we place the cameras at these points (Figure 1). Then, for each camera the same four orientations are defined. These are basically the 45-degree rotations of the negative and positive horizontal and vertical directions (Figure 4).

In a practical application usually the positions as well as the orientations of the cameras are defined by experts. What is more, there can be considerable overlapping among the cameras in certain areas, whereas in other areas there might be even gaps in the camera coverage. Frequently, the cameras are employed in pairs in such a way that each camera is responsible for monitoring its surrounding environment in a semicircle.

To model this, in the REAL scenario we apply four camera pairs, i.e., eight cameras, positioned along the horizontal line with varying distances (Figure 2). Each camera has four orientations which are taken to be the same for all left- and right-directed cameras, respectively. For specific angles among the orientations refer to Figure 5.

In the simulations the cameras spent five time units at each observation point.

##### Test Results

In all of our tests the first 5 min of the simulations were taken into account. With fixed random seeds we ran the same process for all algorithms. Altogether we examined eight algorithms, of which camera scanning is the first one. The average number of individuals entering the playing field within the 5 min time intervals and the corresponding standard deviations can be found in Table 1. The difference between the figures of the full dataset and of the dataset restricted to the camera-accessible region is due to those intruders who had not reached the latter region when the simulation was stopped.

The results of the simulations for the camera scanning algorithm can be found in Table 2. The numbers clearly show that the ABSTRACT scenario is considerably harder than the REAL setting. It is also obvious that the efficiency of the camera scanning algorithm is significantly better in the camera-accessible region. For example, considering the simplicity of the algorithm 49% average tracking ratio in the REAL scenario is quite a decent performance.

In Figure 6 (Top) one can see a more detailed view of the efficiency of tracking in the REAL camera-accessible case. The tracking ratios are separated into ten equally sized bins, and the relative frequency of intruders whose tracking ratio falls into these bins is depicted on the y-axis. At the bottom, the average time spent by intruders in the camera-accessible region is shown for each tracking ratio bin. The second graph indicates that in general the shorter the time an intruder spends in the camera-accessible region the higher the tracking time ratio tends to be.

#### 4.1.2. Camera Tracking and Scanning

In addition to detection, monitoring the behavior and movement of intruders is equally crucial. The significant increase in the accuracy of object detection algorithms in recent years enables extensive automation of this task. Relying on this we extended the basic scanning algorithm with tracking capabilities. Our goal was not only to pursue intruders, but we also aimed to minimize the cost incurred on cameras associated with surveillance.

To achieve this minimization, at each time step we solve an instance of the minimum-cost assignment problem for bipartite graphs [11]. For this we consider the cameras and the intruder groups, and define a cost (see below) between each intruder group camera pair, where the intruder group is in the detection range of the camera. The goal of the algorithm is to find the mapping between intruder groups and cameras with the minimal cost. In such a mapping each intruder group is assigned to at most one camera and vice versa. Examples can be found in Figure 7.

Note that in our case the bipartite graph is unbalanced, since the number of intruder groups and cameras are not necessarily the same. The problem can be solved in $O(ms+s^{2}\log s)$ time, where *m* is the number edges, while *s* is the minimum of the numbers of cameras and intruder groups [11].

The cost between a camera and an intruder group is defined as follows:(3)C=d+α·θ,
where *d* denotes the distance between the camera and the intruder, while θ is the angular difference between the current orientation of the camera and the orientation which points from the center of the camera to the center of the circle representing the intruder group. α is a weight, a non-negative real number.

In the formula θ represents the cost of camera usage, while we included *d* for the purpose of encouraging the cameras to prioritize closer groups over more distant ones.

We experimented with two different strategies for selecting the values of the α weights. In the first case we simply assigned 1 to all of the weights, whereas in the second case α was set to 1/m, where *m* denotes the number of intruders in the corresponding group. By choosing weights in this manner we intended to stimulate cameras to prefer larger groups.

In addition to the two weight selection schemes, we also explored whether it is worthwhile to filter those groups that are taken into account for being tracked by cameras. By default all groups were included among the possible candidates, (Figure 7a). In an alternative scenario we only considered the largest groups where there were at least as many individuals as the number of cameras. In the case of equal group sizes we considered all groups with the given size. For example, if we had two cameras and groups of sizes 2,4,3,3,3,1, then we would have chosen the groups with sizes 4,3,3,3 (Figure 7a,b).

Considering both the weight selection and group filtering strategies, we tried out four different possibilities. When comparing the results, we examined the average tracking ratio, average coverage, and the ratio of non-observed intruder indicators. It was found that the method of selecting weights does not significantly affect the effectiveness of the approach. However, when filtering groups, the strategy focusing on larger group sizes generally proved to be 3% better than the approach that considered all groups indiscriminately. Accordingly, in all of our tests, including the previous one executed for the camera scanning algorithm, we solved the minimum-cost assignment problem using only the largest groups selected by the method described just now.

##### Test Results

In Table 3 one can see the simulation results for the camera tracking and scanning algorithm. Interestingly, in the ABSTRACT case the average tracking ratio has not improved considerably (2%), in fact, at the camera-accessible region it has even become worse by 1%. Since there is a strong positive correlation between the average tracking ratio and the average coverage, it is not surprising that the average coverage has not become fundamentally better either. On the other hand, the ratio of non-observed intruders has increased by 9% on the whole playing field. At the camera-accessible region, the rise is even sharper 21%. When a camera tracks a group, it cannot simultaneously detect new intruders. As a result, they can more easily bypass the security system without being noticed. The next algorithm, camera tracking with forced scanning, will try to prevent this possibility. The only indicator which has significantly improved in comparison to the previous algorithm is the average camera capacity utilization, which has decreased by 7%.

After realizing that the introduction of tracking has only slightly improved the efficiency in the ABSTRACT case, at first glance it might come as a surprise that in the REAL case all of the indicators have become significantly better. For instance the average tracking ratio has increased by 10 and 14% for the full dataset and the camera-accessible region respectively. The difference is primarily due to the fact that in the REAL case the number of cameras is double compared to the ABSTRACT case. In addition, the path system that is under surveillance is also much simpler in the more realistic scenario.

Altogether, this shows that the benefits of combining tracking with scanning instead of relying on simple scanning largely depend on the number of available cameras and the underlying path system.

In Figure 8, one can see the number of covered and non-covered intruders per second. During the calculation, only those intruders were considered who spent exactly 27 s in the camera-accessible region. Specifically, most intruders fell within this time frame. The top figure shows data belonging to the camera scanning algorithm and it clearly shows the periodic behavior of this method. Each camera faces in one direction for a fixed time then moves on, regardless of whether it detects intruder groups or not. The bottom figure belongs to the camera tracking and scanning method. The advantages of implementing tracking are clearly visible in the diagram: if a camera detects an intruder group, it usually does not lose sight of it, thus the number of tracked intruders continues to grow until the optimal state is reached, when the system follows every detectable group. However, the optimal state does not last long. There can be two reasons for this. First, a group may exit the field of view of one camera in such a way that another camera could still see it, but that camera is not currently looking in that direction. Second, it may happen that the algorithm prioritizes tracking a newly arrived group over an already tracked group.

#### 4.1.3. Camera Tracking with Forced Scanning

As the results of the simulations of the camera tracking and scanning algorithm have shown in the ABSTRACT case, while cameras track the already detected groups, they may lose sight of the appearance of new trespassers, thereby the ratio of non-observed intruders may increase significantly. This strategy can also be observed in practice. That is, a smaller group intentionally attracts the attention of cameras and other sensors, while the larger part of the team crosses the border unnoticed.

To address this, in an enhanced version of the algorithm we introduced a time limit regarding how long a camera can track the groups. During this time interval a camera is not necessarily restricted to following only one group. If a camera exceeds the time limit, it must switch to scanning mode and sequentially visit the assigned observation points. After completing this task, it is allowed to start tracking intruders again.

##### Test Results

In the simulations, we set the aforementioned time limit to 20 time steps. The results can be found in Table 4. As was expected, the ratio of non-observed intruders dropped in the ABSTRACT case (8–9%). On the other hand, the average tracking ratio and average coverage values worsened, however, this decrease in efficiency is nearly insignificant 1–2%. Furthermore, the average camera capacity utilization value also increased, still, the algorithm requires only 17% of the overall capacity, which is more than acceptable. All in all, in the ABSTRACT case the introduction of forced scanning significantly improved the efficiency.

On the reverse side of the coin, in the REAL case all of the indicators worsened by 3 to 8%. This shows that the benefits of this enhancement are ambiguous.

### 4.2. UAV Algorithms

#### 4.2.1. UAV Patrolling

Unlike cameras, which can only monitor a restricted region, UAVs are not bound to a specific location. Therefore, the area they can observe is only limited by the lifespan of their batteries. On the other hand, operating them may come with significantly higher costs. The question naturally arises: if we define the patrolling task in a similar way to scanning in the case of cameras, then how do the two systems compare in terms of efficiency?

In a UAV-specific patrol task, instead of defining a fixed set of observation points as with cameras, a route needs to be specified for the UAVs to follow. Inspired by [19,23] to cover a rectangular area, the UAVs are set to fly in a back-and-forth motion along lines perpendicular to a predefined sweep direction. See Figure 9 for an example. It is important to observe that the sweep direction significantly impacts the number of turns taken outside the coverage area, thereby affecting the overall coverage time. Ideally, the optimal sweep direction aligns parallel to the smallest linear dimension of the area [23]. In our case this coincides with the width of the playing field, since it is a rectangle. Note also that the sweeping direction is defined to be opposite to the direction from which the intruders are coming. Along this flying route the UAVs are distributed in such a way that the distance between neighbors is always the same.

##### Test Results

In both the ABSTRACT and REAL cases we employed four UAVs in the simulation. The results can be seen in Table 5. Since a UAV can fly to every location on the playing ground, it is meaningless to restrict the assessment to the camera-accessible region in this case as well as in the case of the next UAV tracking and patrolling algorithm.

A close inspection of the data reveals that the UAV patrolling algorithm is worse than the camera scanning algorithm in every aspect, especially in the REAL case. This is not surprising given that the number of cameras is twice the number of UAVs in this scenario. However, even in the ABSTRACT setting, where the number of cameras and UAVs are the same, the ratio of non-observed intruders jumps from 31% to 41%. Meanwhile, the UAVs worked almost at full capacity, whereas the utilization was only 19% for the cameras. These observations clearly show the disadvantages of applying UAVs for simple, patrolling purposes.

#### 4.2.2. UAV Tracking and Patrolling

As in the case of camera scanning, patrolling can be extended with tracking. In a similar manner as there, in order to assign UAVs to groups to track in a cost-effective way, at every time step an instance of the minimum-cost assignment problem for bipartite graphs should be solved. In this scenario for a UAV, intruder group pair (u,g) the cost is defined as the distance between *u* and *g*, which, more precisely formulated, is the distance between the centers of the circles representing *u* and *g*, respectively. If a UAV does not have any group to track, then it returns to its patrolling activity.

##### Test Results

The simulation results for the UAV tracking and patrolling algorithm can be found in Table 6. It turns out that the average tracking ratio and the average coverage have become better in comparison to the UAV patrolling algorithm in both scenarios. In the REAL case, the improvement, 13%, is quite significant. Meanwhile, the ratio of non-observed intruders has increased by 11% in the ABSTRACT setting, while it has decreased by 11% in the REAL scenario. The ratio of UAV capacity utilization has also considerably decreased in both situations by 20% and 14%.

Putting everything together, the enhancement of the UAV patrolling algorithm with tracking seems to be worthwhile. On the other hand, the camera tracking and scanning algorithm still outperforms its UAV counterpart in every aspect. This, together with the previous results for the simple UAV patrolling algorithm, shows that it is not beneficial to use UAVs for patrolling and tracking purposes alone. Nevertheless, their combination with cameras seems to be promising.

### 4.3. Cameras and UAVs Combined

Drawing conclusions from the results of the previous two algorithms that exclusively use UAVs, in the method combining cameras and UAVs we employ UAVs solely for tracking, while cameras serve both scanning and tracking purposes.

#### 4.3.1. Combination without Cooperation

In the first variant of the method, the camera and UAV system work independently. In concrete terms this means that in each time step we first run a slight modification of the previous UAV tracking and patrolling algorithm in which if a UAV does not have any group to track, then to save energy or to recharge it returns to its original takeoff point. Next, the camera tracking and forced scanning algorithm is also executed. Since there is no communication between the two algorithms it can easily happen that a group is tracked by both a camera and a UAV at the same time. Of course, this is far from being optimal. Still, it might make sense to first assess the efficiency of this rudimentary approach to be able to see the extent of improvement when cooperation is introduced.

##### Test Results

The result of the simulation for the non-cooperative camera, UAV algorithm can be read in Table 7. Even a superficial review of the data reveals the apparent benefits of applying UAVs together with cameras. In comparison to the camera tracking and forced scanning algorithm, average tracking ratio, average coverage, and ratio of non-observed intruders have all improved in the ABSTRACT case by 6–16%. Only average camera capacity utilization has become worse by 2%, which is more than acceptable since it is still below 20%. In the REAL case, the improvement is even more remarkable for the average tracking ratio and the average coverage indicators. What is more, despite the fact that the results for the ratio of non-observed intruders were already quite good for the camera tracking and forced scanning algorithm (6%), the improvement is still observable here (2%). Regarding the UAV tracking and patrolling algorithm, the UAV average capacity utilization values have also improved by 3%.

#### 4.3.2. Combination with Cooperation

It is very easy to introduce cooperation between the UAV and camera phases of the previously defined combined method. Simply, after executing the UAV tracking part, an intruder group that is assigned to a UAV and is located within the detection range of this UAV is not considered as a candidate to be tracked by a camera. In this way we ensure that a group can never be tracked simultaneously by both a camera and a UAV.

Note that we assign the intruder groups to UAVs first because we want to preserve the cameras as much as possible for scanning.

##### Test Results

For the results of the simulation of the cooperative camera, UAV algorithm check Table 8. If one compares the figures with the data of the non-cooperative camera, UAV algorithm, it turns out that in terms of the average tracking ratio and the average coverage indicators, the cooperative version is superior to the non-cooperative one. When the whole playing field is taken into consideration this difference is rather subtle (2–3%), however, in the camera-accessible region it becomes more substantial (7–8%). When it comes to the ratio of non-observed intruders in three out of four cases the non-cooperative version slightly outperforms the cooperative variant by 0.5–3%. In the camera-accessible region of the ABSTRACT simulation, the cooperative variant achieves a better result by 3%. Note though that all of these latter differences seem to be rather insignificant.

In order to be able to contrast the performance of the simplest algorithm, camera scanning, with the most complex one, the cooperative camera, UAV algorithm, consider Table 9. Here, one can see that the average tracking ratio and the average coverage values of the latter algorithm are much better (10–28%). Similarly, the ratio of non-observed intruders has also significantly improved by 11% and 8% for the ABSTRACT and the REAL cases, respectively. Furthermore, the differences between the average camera capacity utilization values seem to be rather irrelevant.

In Figure 10, similar diagrams to that of Figure 8 can be seen. The depicted data belong to the combined algorithm with coordination. The top figure represents the entire playing field, while the bottom represents the camera-accessible region. Comparing the bottom graphs of Figure 8 and Figure 10, the advantages of using UAVs are clearly visible. In this case, before reaching the optimal state the number of tracked intruders increases in a similar fashion, which shows that UAVs do not play an important role in this phase of tracking. However, after reaching the optimal state, the number of lost groups from sight becomes lower. In the top diagram one can also see that if a group leaves the camera-accessible region the UAVs can still effectively track its members. Overall, these observations indicate that UAVs provide the greatest assistance in tracking intruders who have already moved beyond the range of cameras.

## 5. RL Approach

In previous methods, we manually set how much time each camera spends on scanning and how much on tracking. In our next algorithm we let the system learn these ratios. To achieve this we used a solution based on Multi-Agent Reinforcement Learning (MARL), where each camera acts as an individual agent. These camera agents work together with nearby cameras to achieve as much area coverage as possible while minimizing unnecessary movements.

### 5.1. Environment

We have created our custom MARL environment utilizing the PettingZoo API [28], which offers an interface similar to the popular Gymnasium API [29] but is also applicable to MARL problems. An important aspect of our environment is its partially observable nature, as the MARL agent makes decisions based on what the cameras observe at any given time. This feature adds a layer of realism to our simulation, making it more accurate compared to other methods [9].

#### 5.1.1. Observation

The observation for each agent is composed of the camera and intruder positions to compose a $400+N_{\mathrm{camera}}*3$-length vector in the following way: the first 400 places in the observation vector are for up to 200 intruders, where each intruder is described by their x and y coordinates within the camera’s coordinate system. This vector contains only the intruders that are visible by the agent’s camera. Other positions are zeroed out. This makes this problem a partially observable MARL. The other $N_{\mathrm{camera}}*3$ positions in the observation vector are for the x, y coordinates and the pan value of the orientation of the agent’s camera and the two neighboring cameras. All values in the observation vector are normalized relative to the size of the environment. Additionally, this vector is further concatenated with an agent indicator, which is a one hot-encoded index vector that makes the cameras distinguishable.

To deal with the partially observable nature of the problem, we applied frame stacking to the observation with three frames. This means that the current observation is stacked with two preceding states. To add the latter two techniques to the environment, we used the SuperSuit package [30], which contains wrappers for common MARL environment modifiers.

#### 5.1.2. Reward

Our main goal is to maximize the number of detected intruders. This is achieved by assigning a coverage-based reward, denoted as $r_{\mathrm{cov}}^{i}$, to each agent (camera) *i*. This reward is calculated by comparing the number of intruders detected by agent *i*, $N_{\mathrm{observed}}^{i}$, against the total number of intruders that could potentially be detected by the agent given its detection range, $N_{\mathrm{potential}}^{i}$:(4)$$r_{\mathrm{cov}}^{i}=\frac{N_{\mathrm{observed}}^{i}}{N_{\mathrm{potential}}^{i}}$$

To discourage unnecessary movements of the cameras, we introduce a movement penalty for each camera $r_{move}^{i}=\gamma \cdot \theta^{i}$, where $\theta^{i}$ is the rotation angle of the camera, and coefficient γ determines the severity of the penalty. This gives a local reward for each agent:(5)$$r_{\mathrm{local}}^{i}=r_{\mathrm{cov}}^{i}-r_{move}^{i}$$
For our experiments, γ is set to 0.004 and 0.005 in the ABSTRACT and REAL environments, respectively.

To address the challenge of credit assignment in MARL, where individual actions contribute to collective outcomes, we incorporate both local and global reward components similar to [31]. The global coverage reward indicates the collective performance of all agents:(6)$$r_{\mathrm{global}\;\mathrm{cov}}=\frac{\sum N_{\mathrm{observed}}^{i}}{\sum N_{\mathrm{potential}}^{i}}$$

Finally, the total reward for each agent at each time step combines the local and global rewards, weighted to emphasize individual contributions while still valuing collective success:(7)$$r^{i}=0.8\cdot r_{\mathrm{local}}^{i}+0.2\cdot r_{\mathrm{global}}$$

Here, 80% of the agent’s total reward is derived from its own actions (local reward) and 20% reflects the overall system performance (global coverage reward).

#### 5.1.3. Action Space

We specified a simple three-element discrete action space (turn left, no-op, turn right) representing the possible movements of the cameras. To simplify the action space, zoom levels were always set to their maximum settings eliminating the need for zoom control.

### 5.2. RL Algorithm

As the environment is compatible with PettingZoo API, we could utilize Stable Baselines 3 (SB3) [32], a popular RL library with implementations of the most common RL algorithms. We applied a Proximal Policy Optimization agent [33] one of the most widespread RL algorithms. It is a policy gradient method that employs low policy updates. To deal with the multi-agent coordination problem, we incorporated the Parameter Sharing technique. This approach trains a single model that learns an individual policy for each agent with the help of an agent indicator. This simple method has proven effective for solving cooperative MARL problems [34]. We stuck to the default hyperparameters of SB3 throughout our training process.

Figure 11 shows the episode rewards averaged over 100 episodes during training in the ABSTRACT and REAL environments, respectively. It can be observed that the reward values improved during training in both cases.

### 5.3. Test Results

The results of the tests of the RL algorithm can be found in Table 10. Since the algorithm does not handle UAVs, it seems natural to compare its efficiency with the most complex camera method, namely the tracking and forced scanning algorithm. In terms of the average tracking ratio and average coverage, the RL algorithm dominates the other algorithm in the ABSTRACT simulation by 4% and 5% on the whole playing field and by 13% in the camera-accessible region. The relationship reverses in the REAL case, the tracking and forced scanning algorithm performs better by 8% on the whole playing ground and by 15%, 12% at the camera-restricted region. For the ratio of non-observed intruders the situation is similar. The RL algorithm is better in the ABSTRACT setting by 8%, while in the REAL scenario it is substantially weaker by 20%. This last observation might imply that the penalty for letting intruders cross the field without notification should be slightly increased. These results also show that the REAL environment is harder to learn than the ABSTRACT one.

By taking a glimpse at the camera utilization data, it appears that the dominance of the RL algorithm in the ABSTRACT scenario is largely due to a significantly more intense usage of cameras, the difference being 14%. This might show that for the camera tracking and forced scanning algorithm the time spent on the different observation points could be decreased, as well as the time limit after which the camera switches back from tracking to scanning mode.

## 6. Discussion

In our research we investigated the complex characteristics of surveillance methods employed by hybrid monitoring systems. We developed an algorithm based on cameras and UAVs, which effectively detects and then tracks new intruder groups attempting to cross the barrier without authorization. During the construction of our method, we examined seven intermediate algorithms, gaining valuable insights into how various components of these procedures are interconnected and collaborate. To carry out these assessments we designed two different 2D simulation environments: one more theoretical and one closer to reality. The framework we used was developed by an external party.

We started our analysis with a basic camera-based scanning algorithm which simply observed predefined points one after the other. It turned out that adding tracking in order to be able to assess the temporal movement of intruder groups did not necessarily achieve the desired goal. The benefits of this addition largely depend on the number of superfluous cameras and the underlying path system that the intruders prefer to use in their attempts to beat the surveillance system.

Another potential weakness of this extension lies in the fact that while cameras are monitoring previously identified groups, they might fail to notice the arrival of new intruders. This drawback was addressed by forcing the cameras to return to scanning after tracking groups for a certain amount of time. However, the advantage of this enhancement remained unclear.

While developing increasingly complex algorithms to overcome the challenges associated with the task, by leveraging RL methods we also gave the system the chance to find the optimal solution on its own. It turned out that in this case, the realistic scenario is significantly more difficult to learn than the more theoretical one. Still, while in the realistic setting our heuristically built approach outperformed the RL-based algorithm, under the theoretical scenario the latter method proved to be more efficient, offering some insights into how the performance of its opponent algorithm could be improved.

In a similar way, we investigated algorithms that exclusively rely on UAVs. However, it has become evident that employing UAVs for patrolling and attempting to discover new intruders in this way is less effective than the aforementioned basic camera scanning algorithm.

Thus, in the final algorithm, which combines the two sensor types, UAVs are solely responsible for tracking intruder groups, while cameras can perform both tracking and scanning tasks depending on the circumstances. Furthermore, in order to minimize the costs of sensor usage in each time step, two instances of the linear sum assignment problem are solved.

## Figures and Tables

**Figure 1 sensors-24-04490-f001:**
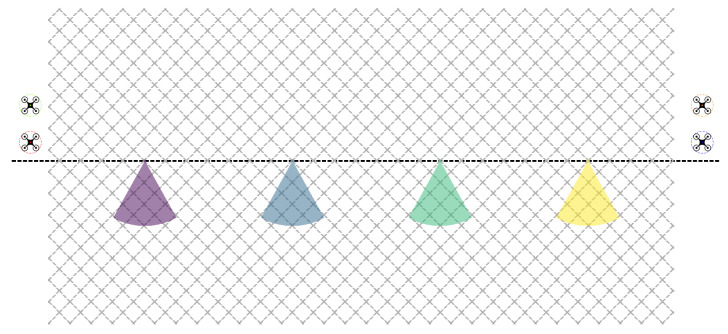
The ABSTRACT environment.

**Figure 2 sensors-24-04490-f002:**
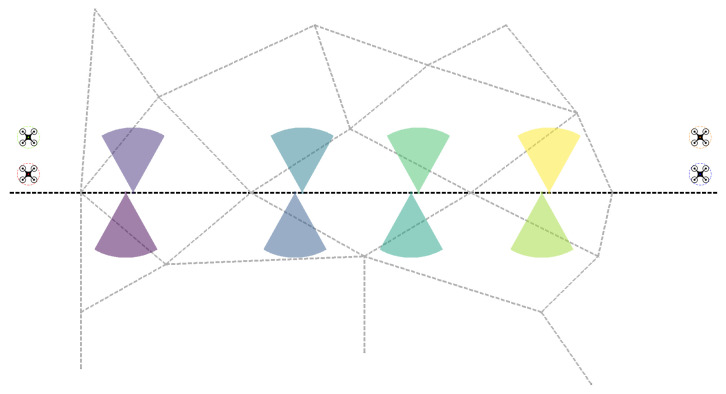
The REAL environment.

**Figure 3 sensors-24-04490-f003:**
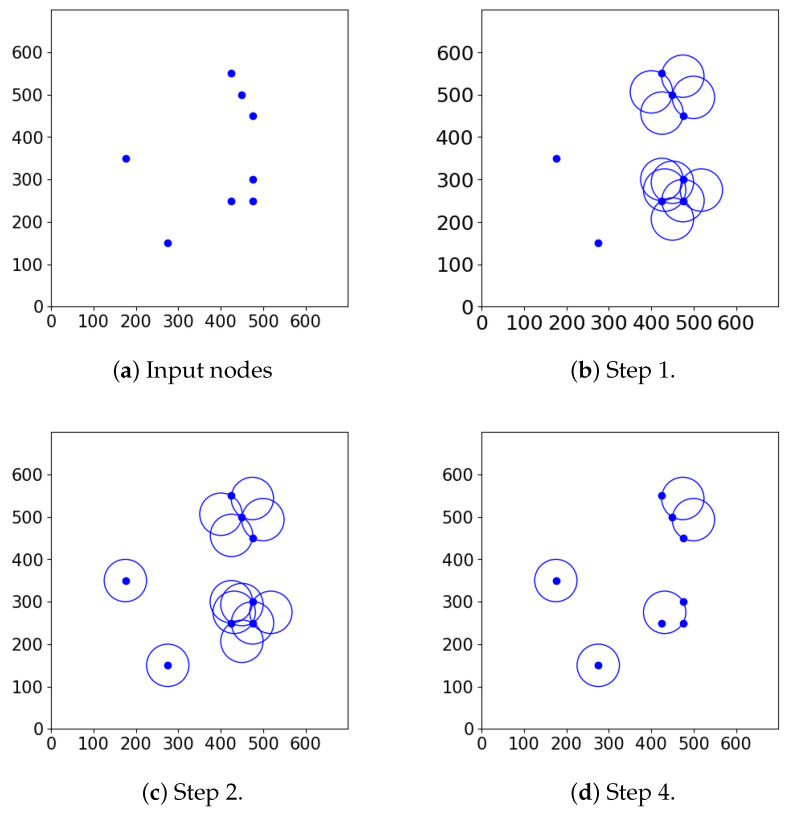
Visualization of the minimum circle cover algorithm. Note that Step 3 is not depicted.

**Figure 4 sensors-24-04490-f004:**
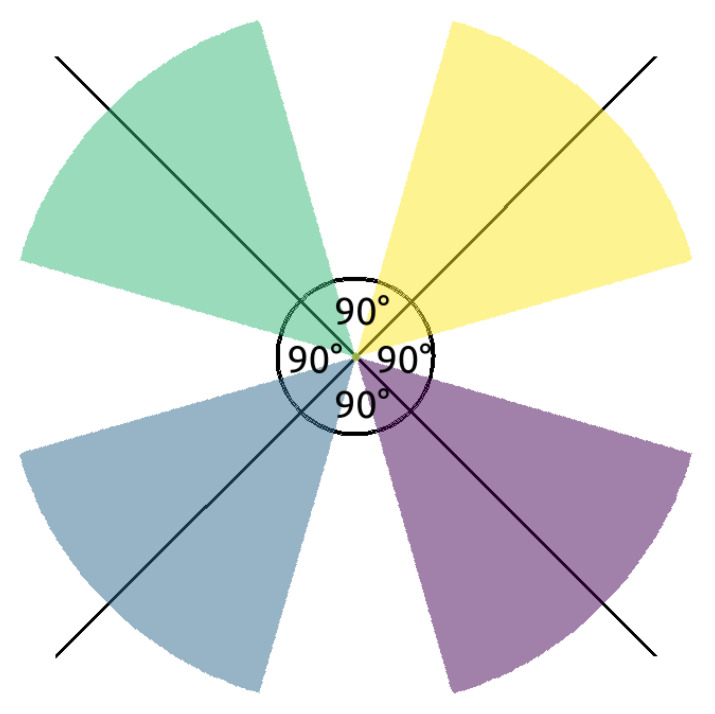
Abstract camera pattern.

**Figure 5 sensors-24-04490-f005:**
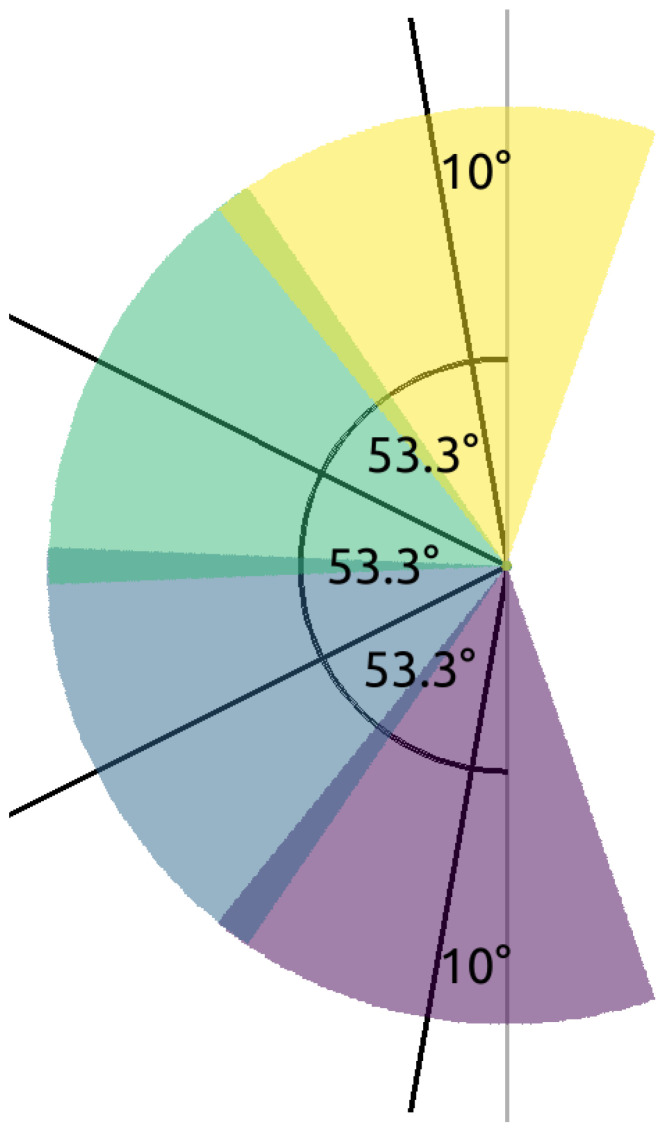
Real camera pattern.

**Figure 6 sensors-24-04490-f006:**
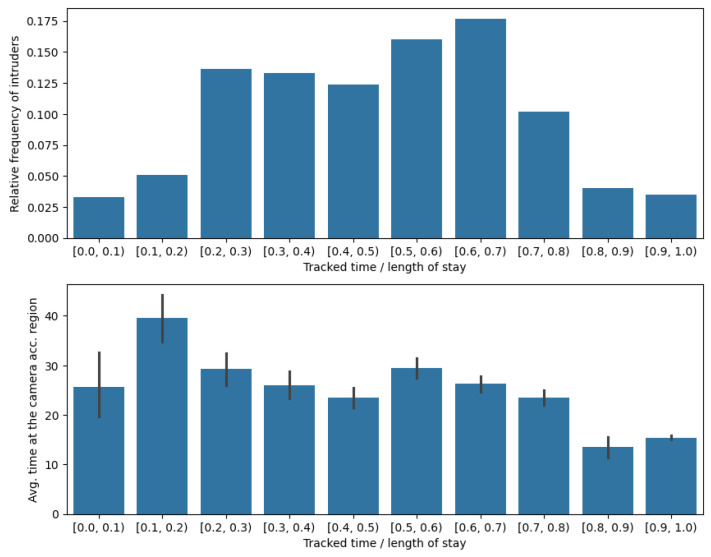
Tracking ratio at the REAL, camera-accessible region. (**Top**) relative frequency of intruders in tracking ratio intervals. (**Bottom**) average time spent at the camera-accessible region by intruders belonging to different tracking ratio intervals.

**Figure 7 sensors-24-04490-f007:**
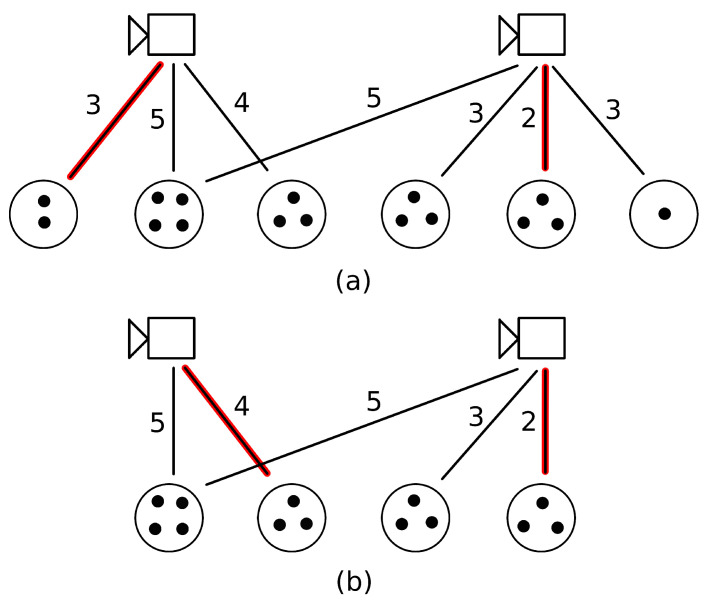
Examples for the minimum-cost assignment problem for bipartite graphs. The black dots in an intruder group represented by a circle denote the number of intruders in the group. The label of an edge between a camera and an intruder group stands for the cost. The red highlighted edges give a minimal cost assignment. (**a**) The scenario in which all intruder groups are considered. (**b**) The situation when only the largest intruder groups are taken into account.

**Figure 8 sensors-24-04490-f008:**
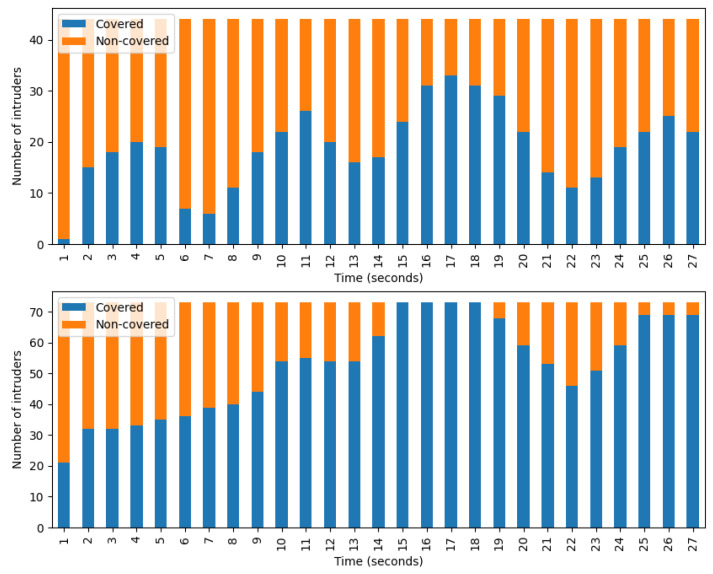
Number of covered and non-covered intruders per second who stayed for 27 s at the REAL, camera-accessible region. (**Top**) camera scanning. (**Bottom**) camera tracking and scanning.

**Figure 9 sensors-24-04490-f009:**
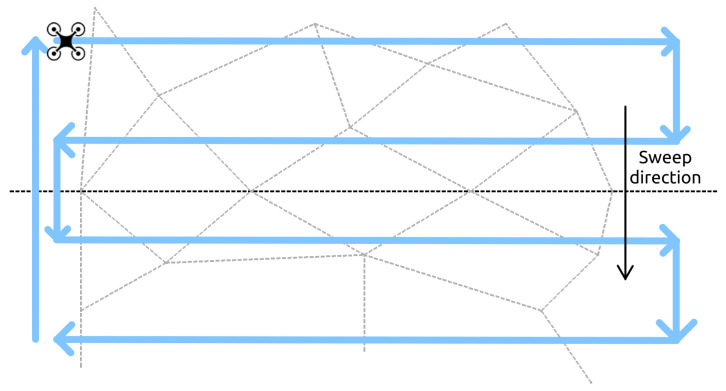
UAV patrol pattern.

**Figure 10 sensors-24-04490-f010:**
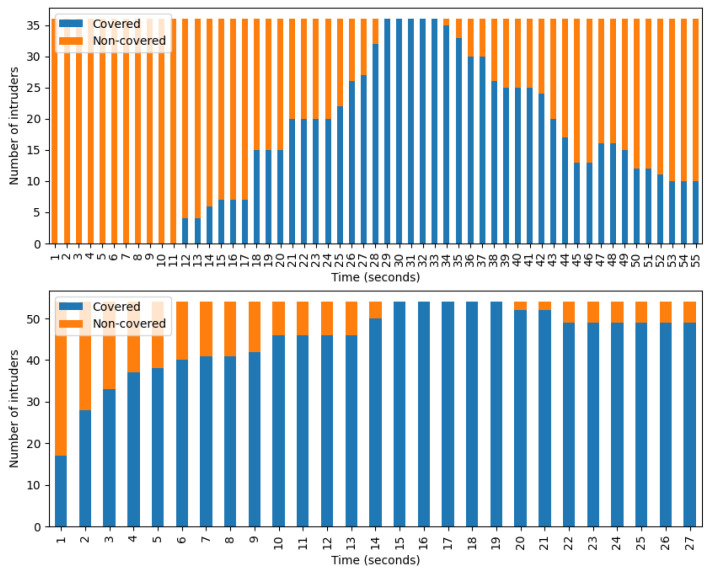
The number of covered and non-covered intruders per second. Both diagrams depict data belonging to the combined algorithm with coordination. (**Top**) entire playing field. (**Bottom**) camera-accessible region.

**Figure 11 sensors-24-04490-f011:**
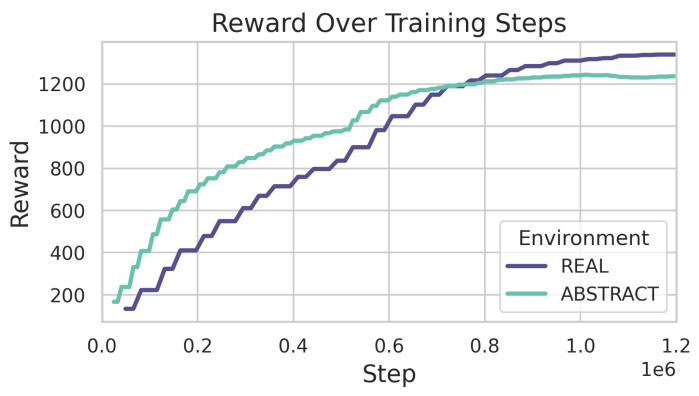
Rl training.

**Table 1 sensors-24-04490-t001:** Average number of intruders entering the playing fields and the corresponding standard deviations in the different scenarios.

	ABSTRACT		REAL	
No. of Intruders	Full	Camera Acc.	Full	Camera Acc.
Average	1695.75	1473.1667	582.375	564.5
Standard deviation	118.3499	165.1821	18.2821	20.2382

**Table 2 sensors-24-04490-t002:** Simulation results for the camera scanning algorithm.

	ABSTRACT		REAL	
Indicator	Full	Camera Acc.	Full	Camera Acc.
Average tracking ratio	0.095	0.3063	0.1969	0.4936
Average coverage	0.1279	0.3195	0.2323	0.5228
Non-observed ratio	0.3096	–	0.1248	–
Average camera utilization	0.1917	0.1917	0.1159	0.1159

**Table 3 sensors-24-04490-t003:** Simulation results for the camera tracking and scanning algorithm.

	ABSTRACT		REAL	
Indicator	Full	Camera Acc.	Full	Camera Acc.
Average tracking ratio	0.1151	0.292	0.3007	0.6359
Average coverage	0.1416	0.3563	0.328	0.7104
Non-observed ratio	0.3956	–	0.036	–
Average camera utilization	0.125	0.125	0.1312	0.1312

**Table 4 sensors-24-04490-t004:** Simulation results for the camera tracking and forced scanning algorithm.

	ABSTRACT		REAL	
Indicator	Full	Camera Acc.	Full	Camera Acc.
Average tracking ratio	0.105	0.2757	0.2645	0.5682
Average coverage	0.1312	0.3502	0.2938	0.6269
Non-observed ratio	0.3133	–	0.06	–
Average camera utilization	0.1743	0.1743	0.1321	0.1321

**Table 5 sensors-24-04490-t005:** Simulation results for the UAV patrolling algorithm.

	ABSTRACT		REAL	
Indicator	Full	Camera Acc.	Full	Camera Acc.
Average tracking ratio	0.0809	–	0.077	–
Average coverage	0.0938	–	0.0844	–
Non-observed ratio	0.4145	–	0.4768	–
Average UAV utilization	0.9229	–	0.945	–

**Table 6 sensors-24-04490-t006:** Simulation results for the UAV tracking and patrolling algorithm.

	ABSTRACT		REAL	
Indicator	Full	Camera Acc.	Full	Camera Acc.
Average tracking ratio	0.1146	–	0.2055	–
Average coverage	0.1377	–	0.2127	–
Non-observed ratio	0.5238	–	0.3688	–
Average UAV utilization	0.7279	–	0.8019	–

**Table 7 sensors-24-04490-t007:** Simulation results for the non-cooperative camera, UAV algorithm.

	ABSTRACT		REAL	
Indicator	Full	Camera Acc.	Full	Camera Acc.
Average tracking ratio	0.2241	0.3354	0.4343	0.6128
Average coverage	0.2873	0.441	0.478	0.6912
Non-observed ratio	0.1969	–	0.0223	–
Average camera utilization	0.1901	0.1901	0.14	0.14
Average UAV utilization	0.6985	0.6985	0.7673	0.7673

**Table 8 sensors-24-04490-t008:** Simulation results for the combined camera, UAV algorithm.

	ABSTRACT		REAL	
Indicator	Full	Camera Acc.	Full	Camera Acc.
Average tracking ratio	0.2402	0.403	0.4627	0.6892
Average coverage	0.3024	0.5134	0.5084	0.767
Non-observed ratio	0.2022	–	0.0497	–
Average camera utilization	0.1783	0.1783	0.1418	0.1418
Average UAV utilization	0.7283	0.7283	0.7637	0.7637

**Table 9 sensors-24-04490-t009:** Difference between the simulation results for the combined camera, UAV, and the camera scanning algorithms.

	ABSTRACT		REAL	
Indicator	Full	Camera Acc.	Full	Camera Acc.
Average tracking ratio	0.1452	0.0967	0.2658	0.1956
Average coverage	0.1745	0.1939	0.2761	0.2442
Non-observed ratio	−0.1074	–	−0.0751	–
Average camera utilization	−0.0134	−0.0134	0.0259	0.0259

**Table 10 sensors-24-04490-t010:** Simulation results for the RL algorithm.

	ABSTRACT		REAL	
Indicator	Full	Camera Acc.	Full	Camera Acc.
Average tracking ratio	0.1497	0.4105	0.185	0.417
Average coverage	0.1879	0.482	0.2157	0.5073
Non-observed ratio	0.2312	–	0.2702	–
Average camera utilization	0.3128	0.3128	0.1018	0.1018

## Data Availability

The algorithms described in the article are parts of a larger program; however, for business reasons, we do not make the entire system open-source.

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
