# Peer review of "Comparison of Barrier Surveillance Algorithms for Directional Sensors and UAVs"

_sensors, 2024, doi:10.3390/s24144490_

Round 1

Reviewer 1 Report

Comments and Suggestions for Authors

Dear authors, 

thank you for addressing very interesting topic. For future reference, consider adding paragraphs, figures and graphs showing the tracking performance of your algorithms. From this perspective, the work deserves some improvements. Target assignment strategy is also not transparently explained. 

Best regards

Author Response

Comments 1: For future reference, consider adding paragraphs, figures and graphs showing the tracking performance of your algorithms. 

Response 1: Thank you for pointing this out. We added two figures and additional explanations (highlighted in yellow) regarding the tracking behavior of the algorithms. The first figure and the related description can be found at the end of Section 4.1.2 (page 14-15). The second figure and the related explanation is at the end of Section 4.3.2 (page 19-20).

Comments 2: Target assignment strategy is also not transparently explained.

Response 2: We agree with this comment. Therefore, we added a figure to further clarify the explanation. It is in Section 4.1.2 page 13. The time complexity of the algorithm has also been made more precise (page 13, highlighted yellow). The original two separate references (see below) have been changed to a better one. Basically, the new reference is an abbreviated version of the original first reference.

Original references:

Ramshaw, L.; Tarjan, R.E. On minimum-cost assignments in unbalanced bipartite graphs. HP Labs, Palo Alto, CA, USA, Tech. Rep. HPL-2012-40R1 2012, 20.

Lawler, E.L. Combinatorial optimization: Networks and matroids. Bull. Amer. Math. Soc 1978, 84(3), 461--463.

New reference:

Ramshaw, L.; Tarjan, R.E. A weight-scaling algorithm for min-cost imperfect matchings in bipartite graphs. In Proceedings of the 2012 IEEE 53rd Annual Symposium on Foundations of Computer Science, 2012; pp. 581--590.

Reviewer 2 Report

Comments and Suggestions for Authors

Border surveillance and the monitoring is an important research topic. However, the manuscript has several problems:

1. The research was done using only simulation data, which cannot demonstrate the effectiveness of the proposed method in real scenarios.

2. The proposed method was not compared with advanced methods in literatures.

3. The literature review does not cover enough SOTA methods. Only twelve papers were reviewed in the Introduction section.

Comments on the Quality of English Language

There are some typos.

Line 11-12: As a result we have gained valuable insights into how various components of 11 these algorithms are interconnected and “collaborate”.

Author Response

Comments 1: The research was done using only simulation data, which cannot demonstrate the effectiveness of the proposed method in real scenarios.

Response 1: Basically, we agree with this comment. However, it is very difficult if not impossible to find real-world data with which the effectiveness of intruder detection algorithms could be tested. Such data should include the trajectories of numerous real intruders in a region monitored by a sensor network. We are not aware of any publicly available dataset that would contain this type of information. All of the papers we read through used simulated data to represent the movement of intruders. Of course, if such a dataset existed, we would happily use it to test the performance of the developed algorithms.

Comments 2: The proposed method was not compared with advanced methods in literatures.

Response 2: Thank you for pointing this out, we strongly agree with this comment, therefore we added a new section to our paper listing the related works (Section 2). It is highlighted in yellow and can be found on pages 3 and 4.

Comments 3: The literature review does not cover enough SOTA methods. Only twelve papers were reviewed in the Introduction section.

Response 3: Again, we agree with this comment and intend to address this problem with the added Related works section. In some cases, we could have referred to more recent publications, but we felt that this would not change the essence of our argument. Namely, the differences between the referenced works and our own results would remain the same even in the case of newer publications.

Reviewer 3 Report

Comments and Suggestions for Authors

The proposed method can be implemented when both the number and positions of the targets are unknown. Would there be any performance loss compared with the exiting method applied under the case where this information is known? 
